# Quality considerations and major pitfalls for high throughput DNA-based newborn screening for severe combined immunodeficiency and spinal muscular atrophy

**Jessica Bzdok**[1,2]*, **Ludwig Czibere**[2], **Siegfried Burggraf**[2], **Olfert Landt**[3], **Esther M. Maier**[2], **Wulf Röschinger**[2], **Michael H. Albert**[4], **Sebastian Hegert**[5], **Nils Janzen**[5,6,7], **Marc Becker**[1,2], **Jürgen Durner**[1,2]

1 Department of Operative/Restorative Dentistry, Periodontology and Pedodontics, Ludwig-Maximilians-Universität München, Munich, Germany, 2 Laboratory Becker MVZ GbR, Munich, Germany, 3 TIB Molbiol Syntheselabor GmbH, Berlin, Germany, 4 Department of Paediatrics, Dr. von Hauner Children's Hospital, Ludwig-Maximilians-Universität München, Munich, Germany, 5 Screening-Labor Hannover, Hanover, Germany, 6 Department of Clinical Chemistry, Hanover Medical School, Hanover, Germany, 7 Division of Laboratory Medicine, Centre for Children and Adolescents, Kinder- und Jugendkrankenhaus Auf der Bult, Hanover, Germany

* j.bzdok@labor-becker.de

**Data Availability Statement:** All relevant data are within the manuscript and its supporting Information files.

## Abstract

### Background

Many newborn screening programs worldwide have introduced screening for diseases using DNA extracted from dried blood spots (DBS). In Germany, DNA-based assays are currently used to screen for severe combined immunodeficiency (SCID), spinal muscular atrophy (SMA), and sickle cell disease (SCD).

### Methods

This study analysed the impact of pre-analytic DNA carry-over in sample preparation on the outcome of DNA-based newborn screening for SCID and SMA and compared the efficacy of rapid extraction versus automated protocols. Additionally, the distribution of T cell receptor excision circles (TREC) on DBS cards, commonly used for routine newborn screening, was determined.

### Results

Contaminations from the punching procedure were detected in the SCID and SMA assays in all experimental setups tested. However, a careful evaluation of a cut-off allowed for a clear separation of true positive polymerase chain reaction (PCR) amplifications. Our rapid in-house extraction protocol produced similar amounts compared to automated commercial systems. Therefore, it can be used for reliable DNA-based screening. Additionally, the

**Funding:** The author(s) received no specific funding for this work.

**Competing interests:** The authors have declared that no competing interests exist.

amount of extracted DNA significantly differs depending on the location of punching within a DBS.

## Conclusions

Newborn screening for SMA and SCID can be performed reliably. It is crucial to ensure that affected newborns are not overlooked. Therefore a carefully consideration of potential contaminating factors and the definition of appropriate cut-offs to minimise the risk of false results are of special concern. It is also important to note that the location of punching plays a pivotal role, and therefore an exact quantification of TREC numbers per µl may not be reliable and should therefore be avoided.

## Introduction

The year 2023 marked the 60[th] anniversary of the first newborn screening (NBS) initiative for phenylketonuria [1], one of the first comprehensive approaches to a preventive strategy for inborn metabolic and endocrine disorders in modern medicine, and the foundation for the most successful secondary prevention measure. In recent decades, NBS has relied on the determination of biochemical parameters such as hormones, metabolites (amino acids, acyl carnitines) and enzyme activities to detect mainly endocrine disorders or inborn errors of metabolism in a pre-symptomatic state. Nowadays over 50 inborn errors can be detected from dried blood spots (DBS) punched from Guthrie filter cards [2, 3] with further potential applications [4]. In recent years, however, treatment options have become available for several diseases without a specific biochemical parameter. For these disorders, the detection of alterations at the DNA level offers a perspective for inclusion in NBS programs, ensuring early diagnosis. In addition, recent studies have highlighted the benefits of early diagnosis for existing treatments, such as human stem cell transplantation (HSCT), leading to increased survival rates [5].

Spinal muscular atrophy (SMA) is an autosomal recessive neuromuscular disease with high morbidity and mortality, leading to premature death if untreated. The prognosis for affected newborns has changed dramatically with the development of therapeutic options like Zolgensma® or Spinraza®/Nusinersen [6]. SMA is caused by a homozygous deletion of exon 7 or both exon 7 and exon 8 in the survival of motor neuron 1 *(SMN1)* gene in >95% of cases, easy to target by polymerase chain reaction (PCR) analysis [7, 8].

Severe combined immunodeficiency (SCID) is the collective term used to describe at least 20 known rare genetic disorders in which both T and B cell immunity, components of the adaptive immune system, are deficient [9]. So far, all known gene mutations causing defects in the development of healthy naive T cells or a failure during thymopoiesis lead to the absence or very low numbers of T cells. This results in a combined cellular and humoral immunodeficiency with no or non-functional B cells [10]. Infants with SCID are highly susceptible to severe infections which, if undiagnosed and untreated, lead to a nearly 100% mortality rate within the first two years of life. Early diagnosis and implementation of life-saving therapies are crucial for a positive outcome [11, 12]. Recent studies have shown that the survival rates for SCID can be significantly improved if treatments such as HSCT are applied within the first few months of life, particularly before the first infection [13].

To identify SCID and other forms of T cell lymphopenia (TCL), a DNA-based approach was described by amplifying T cell receptor excision circles (TREC) [14, 15]. TRECs are DNA by-products that originate from the recombination of T cell receptors during thymopoiesis.

Low amounts or absence of TRECs are indicative of TCL. Therefore, neonatal TREC levels determined by TREC specific PCRs can be used to detect impaired T cell development and hence to screen for SCID [16]. If no or few TRECs can be detected, further clarification of the diagnosis is essential.

In 2010, screening for severe combined immunodeficiency (SCID) was introduced in some states of the US [17] as the first DNA-based NBS assay. Since then, the list of DNA-based NBS assays has continued to grow [18].

In Germany, screening for SCID is currently part of the NBS panel since 2019, SMA and sickle cell disease (SCD) have been added since 2021. All three parameters can be tested utilising DNA-based methods.

One of the most widely used methods in molecular diagnostics is PCR. PCR has a long history of use, for example in pathogen detection, and due to the high intrinsic sensitivity of this method, contamination is one of the major problems. This well-known fact is addressed by regulatory guidelines that, for example, do not allow PCR for pathogens from a vial of blood that has already been opened and used for another test, such as a blood cell count [19–21]. The use of PCR on DBS is quite different from samples such as whole blood. In particular, the production of punches from DBS is a process that does not meet the requirements of a diagnostic PCR assay. The punching process creates dust particles containing blood cells that may contaminate neighbouring vials [22].

For genetic assays based on PCR the problem of these contaminations is likely to appear in mutation/disease specific real-time PCRs (qPCR) that rely on the presence or absence of amplification signals as used for SMA [7], cystinosis [23], SCD [24] or SCID. Especially assays where the disease is detected by a failure to amplify DNA might be prone to false results due to contamination, i.e. an amplification signal can be detected that would exclude a disease state. Therefore, we decided to examine the assays for SMA and SCID for this potential problem.

Commonly used DBS punches range from 1.5 mm to 3.2 mm. The 3.2 mm punch equals approximately 3 μl of blood [25, 26]. Thus, the amount of nucleic acid contained in each sample is limited. In comparison to SMA screening, which targets whole genomic DNA, the detection of TREC is more challenging. Even in healthy individuals the copy numbers of TREC are relatively low compared to genomic DNA [27].

Czibere *et al.* showed a simple, highly cost and time efficient procedure for nucleic acid extraction combined with qPCR on 384-well plates allowing high throughput NBS for SMA [7]. Meanwhile, up to three parameters are routinely analysed in the German Newborn screening program using DNA-based techniques [7, 24, 28]. All published assays highlight the benefits of DNA-based NBS. To our knowledge, there are no publications that address the risk of false negative results.

The aim of our study was to (i) examine the impact of pre-analytic DNA carry-over, (ii) compare the robustness of distinguishing true PCR signals from contamination for *SMN1* deletion and TREC (iii) to analyse the impact of punching location on DBS and (iv) to assess the feasibility of determining precise TREC numbers from DBS.

## Material and methods

### Preparation of TREC negative and normalised TREC positive peripheral blood

For this study, only plasmid material or archived human samples have been used. Only age related data have been accessed prior to any testing between 3/2/2021 and 3/3/2021. No individuals could be identified during or after data collection by the authors, as the samples were pooled prior to testing.

The number of TRECs in peripheral blood decreases as thymopoiesis declines with age. To obtain a larger volume of TREC negative blood, peripheral whole blood samples anticoagulated with EDTA were pooled from anonymised donors over 80 years old. DNA from 200 µl was extracted on the MagNA Pure 96 instrument (Roche, Mannheim, Germany) using the MagNA Pure 96 DNA and Viral NA Small Volume Kit and following the Pathogen Universal 200 v. 4.0 protocol with 50 µl elution volume. PCR was performed with LightMix® KIT TREC SMA HBB Newborn (Roche) according to the manufacturer's protocol to confirm if the pooled samples gave negative results for TREC. From several different pools, only one pool was negative for TREC (TREC negative blood; TREC-NB). TREC-NB was used as diluent in the following experiments to maintain the (genomic) *SMN1* gene as a reference within the same range, while the TREC concentration was reduced.

As the second step, peripheral whole blood samples anticoagulated with EDTA from anonymised newborns were pooled and tested with the LightMix® KIT Newborn. For reference of the quantification, a serial dilution of TREC plasmid DNA provided by TIB Molbiol (Berlin, Germany) was used. The pooled blood from newborns was diluted with the TREC-NB to approximately 300 TREC/µl (= normalised TREC positive blood; TREC-PB). In addition, serial dilutions were performed on the TREC-PB using the TREC-NB as a diluent. The dilutions were made in 5 steps (1:2; 1:5; 1:10; 1:100; 1:1,000), corresponding to approximately 300 to 0.3 copies/µl.

## Preparation of DBS

Blood samples were manually pipetted (80 µl per spot) onto filter cards (TFN specimen collection card, Ahlstrom-Munksjö, Bärenstein, Germany) and air-dried for at least 48 h. 3.2 mm punches were collected in 96-well PCR plates (04-083-0150, nerbe plus GmbH & Co. KG, Winsen/Luhe, Germany) using an automated Panthera Puncher 9 System (PerkinElmer, Waltham, MA, USA).

## Rapid nucleic acid extraction from DBS

Nucleic acid extraction from DBS was performed as published previously as CXCE-buffer extraction [7], including modifications as described below. Throughout washing and extraction steps, pipetting was performed using the ViaFlo96 system (Integra Biosciences, Zizers, Switzerland). For rehydration DBS punches were incubated in 96-well plates in 50 µl of water for 10 min on a plate shaker at 200 rpm (IKA Labortechnik, Staufen, Germany). 150 µl of CX-buffer (1× PBS and 0.5% Thesit®) was added and incubated for further 10 min on the plate shaker at ~200 rpm. After centrifugation (Rotanta 460, Hettich GmbH, Tuttlingen, Germany) at 1,000 rpm for 2 min, the supernatant was removed with an eight-channel vacuum device (Vacusafe, Integra). Subsequently, 150 µl of water was added, followed by another centrifugation step for 2 min at 1,000 rpm. Water was removed with the vacuum device and 50 µl of CE-buffer (10 mM Tris, 0.25 mM EDTA, and 2 mM NaOH; pH 11) was added to each well. Plates were sealed using a PCR sealing foil (nerbe plus) and frozen for at least 20 minutes at -70˚C. For thawing and releasing of DNA, plates were placed in a thermal cycler (Applied Biosystems 2720, Foster City, CA) for 10 min at 92 ˚C. The freezing step is not part of the procedure originally described by Czibere *et al.* 2020. To test the efficiency of the freezing procedure, samples were taken from the routine testing as described before [7].

## MagNA Pure96 nucleic acid extraction from DBS

For extraction from DBS three punches were homogenised in 600 µl of PBS using 1.4-mm zirconium beads (Precellys, Bertin Technologies, Montigny le Bretonneux, France) in a Precellys24

homogenizer. 200 μl of the homogenised punches (corresponding to one punch) were transferred to a 96-deepwell plate (Roche, Mannheim, Germany). For nucleic acid isolation, the MagNA Pure 96 DNA and Viral NA Small Volume Kit following the Pathogen Universal 200 4.0 protocol with 50 μl elution volume was used on the MagNA Pure 96 system (Roche).

## qPCR

Vacuum dried LightMix® KIT Newborn (Roche) was dissolved as described in the manufacturer's instructions. 7 μl of reaction mix consisting of 0.5 μl LightMix® KIT Newborn, 2.0 μl Multiplex DNA Master (Roche) and 4.5 μl $H_2O$ were distributed per reaction to a 384-well plate (Roche). 3 μl of eluted DNA was added to the reaction mix using the ViaFlo96 system (Integra) to transfer the DNA elution from the 96-well plates to the 384-well plate. qPCR was performed on a LightCycler480II instrument (Roche) with the following PCR profile: initial denaturation at 95°C for 10 min, followed by 45 cycles in three steps at 95°C for 5 s, 60°C for 10 s, and 72°C for 15 s. A single acquisition of fluorescence signals was included in the 60°C step. A final melting step (95°C 30 s; 45°C 2 min; 75°C continuous fluorescent measurement with a ramping rate of 0.19°C/s) was included in the protocol to differentiate between haemoglobin beta chain (*HBB*) wild- and variant types. The fit point or second derivative maximum analysis of the LightCycler software (Roche) was used to analyse qPCR fluorescence signals. To assure comparability between runs for fit point analysis, the noise band was always set to the identical fixed value and the same standard positive control sample was used throughout the study. Cut-offs for TREC and SMA were determined by subtracting one Cq from the lowest Cq value of the negative samples (i.e. the sample with the highest amount of contaminating DNA), which is equivalent to twice the amount of the highest concentration of contaminating DNA detected in the experiments. A standardised positive control for TREC and a positive sample for SMA were included in each qPCR run. Since the SMA assay and the TREC assay are included in a single reaction the PCR signal from each of the two assays served as internal control for the other assay.

## EnLite

The EnLite Neonatal TREC-KREC kit (4153–0010, PerkinElmer) was performed on 1.5 mm DBS punches in the Screening Laboratory Hanover as described by the manufacturer.

## Comparison of the rapid in-house CXCE extraction *vs.* MagNA Pure 96 extraction *vs.* EnLite

Filter cards (n = 50) prepared with TREC-PB and the previously prepared dilution series of TREC blood (~ 300 to 3 copies/μl; 80 μl per spot) were used for the experiment. Peripheral whole blood (n = 8) samples anticoagulated with EDTA were additionally extracted on the MagNA Pure 96 as a liquid sample (200 μl blood as well as 3 μl blood in 197 μl PBS- buffer equivalent to one DBS punch) to evaluate any differences due the different extraction procedures. The dilution series was run in a 3-fold approach using the rapid CXCE-buffer extraction from the same DBS-cards and spot as the isolation of the DBS punches using the MagNA Pure 96. Additional DBS punches (n = 16) of TREC-PB samples were tested with the EnLite (PerkinElmer) test according to the manufacturer's instructions.

## Evaluation of pre-analytic DNA carry-over

92 DBS were punched on a 96-well plate, 4 positions remained empty for PCR quality controls used at a later stage. The 92 punches consisted of 62 punches from TREC-PB and 10 punches

from empty filter cards and 20 TREC-NB distributed evenly. A second plate was produced the same way. qPCR was performed in duplicates for each plate on a 384-well plate resulting in a total number of 368 reactions. To exclude a methodical bias of the CXCE-buffer extraction with the LightMix® KIT Newborn qPCR assay, the contaminations were also assessed using a completely different setup in the Hanover screening laboratory with the EnLite system. 766 punches from empty filter cards were interspersed between TREC positive samples measured on 10 different days in a routine setting.

## Assessment of DNA distribution on DBS

Filter cards were dripped with the TREC-PB. Two punches were taken from each spot (DBS), one in the middle/inner area (M) and one from the periphery/edge area (P). Analyses were performed in triplicates from each DBS/spot using the rapid CXCE-buffer nucleic acid extraction. Additionally, duplicates were extracted for both areas on the MagNA Pure 96 to exclude method-specific differences. The identical DBS cards were used applying the same approach in the Screening Laboratory Hanover using the EnLite system.

## Ethics statement

The study was conducted in accordance with the Declaration of Helsinki, the study protocol was presented to the Ethics Committee of the Ludwig Maximilians University Munich. The committee issued a letter of no counselling obligation for the study presented here (Reference: 24–0246 KB), stating that if the research is solely conducted on the basis of data and samples that are irreversibly anonymised there are no ethical or legal objections to the project, thus also no informed consent was required.

## Data collection

Age-related data were retrieved anonymously prior to any testing between 3/2/2021 and 3/3/2021, from a statistics tool (Quickstat, medat computersysteme, Munich, Germany). Quantitative data were generated using the LightCycler software (Version 1.5.1.62 SP3) and exported to Microsoft Excel and R [29], respectively. The data were generated between 6/7/2021 to 8/27/2021, 9/1/2022 to 11/25/2022 and 4/3/2023 to 4/28/2023, respectively.

## Statistics

Cq values are presented as means ± 95% confidence intervals and Mann-Whitney-U tests were calculated using R [29].

## Results

### Modified rapid nucleic acid isolation increases DNA yield

To assess whether adding a freezing step to the nucleic acid isolation process would improve the DNA yield, we compared the respective Cq values with and without freezing. Without the freezing step, a mean Cq value of 29.11 ± 0.0457 with 95% confidence was obtained for TREC signals, whereas with the addition of a freezing step, a mean Cq value of 27.59 ± 0.0362 with 95% confidence was observed. For *SMN1*, a mean Cq value of 27.8 ± 0.0392 was detected before adding a freezing step and a Cq of 25.02 ± 0.0359 with 95% confidence after. For both, *SMN1* and TREC the difference was statistically significant at p < 0.001 (Fig 1 and S1 Dataset). The added freezing step indeed resulted in a significant increase in nucleic acid yield and thus in the number of TRECs detected.

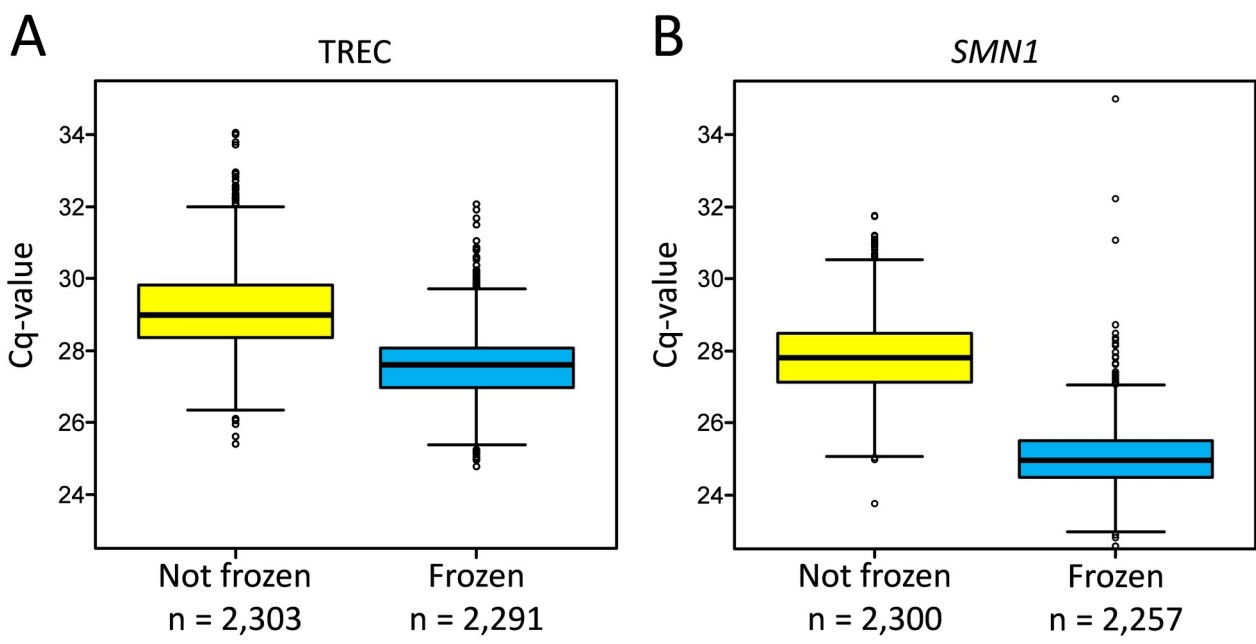

**Fig 1. Effect of freezing in the CXCE-buffer extraction.** Comparison of Cq values without freezing (not frozen, yellow) and with freezing (frozen, blue) in the TREC (A) and the *SMN1* (B) assays.

### DNA yield in rapid CXCE-buffer and MagNA Pure 96 extraction

We compared the CXCE-buffer extraction to automated systems to exclude a methodologic bias in the nucleic acid purification process. There was no significant difference between the modified CXCE-buffer extraction and the MagNA Pure 96 extraction (Fig 2; S2 and S3 Datasets). On average, the modified CXCE-buffer extraction resulted in Cq values of 28.2 ± 0.416, while the MagNA Pure 96 extraction reached Cq values of 28.1 ± 0.608 on average for the normalised TREC-PB. No differences were found between the rapid CXCE-buffer extraction and the MagNA Pure 96 extraction in the additional dilution series tested. The dilution series consisted of 5 steps (1:2; 1:5; 1:10; 1:100; 1:1,000) corresponding to approximately 300 to 0.3 copies/µl. When nucleic acid was extracted using the MagNA Pure 96 method, the results were identical to those obtained using the CXCE-buffer extraction method. The only difference observed was a variation in fluorescence intensity, which could be attributed to residual staining from blood components in the CXCE eluate (Fig 2). Furthermore, the additional samples tested with the EnLite system in a series of 16 measurements showed comparable results for the normalised TREC-PB with a mean value of 287 TREC/µl blood sample (TREC-PB ~ 300 TREC/µl). Taken together, the CXCE-buffer and automated extraction systems yielded comparable amounts of DNA.

### TREC limit of detection

To determine the limit of detection, we analysed a serial dilution of plasmid DNA containing the TREC locus provided by TIB Molbiol. We were able to reliably detect 5 TREC/copies per PCR reaction (i.e. 3 µl plasmid DNA; S3 Dataset and S1 Fig). Thus, 5 copies in 3 µl (1.66 copies/µl) correspond to 83 copies in 50 µl eluate from one DBS punch. The serial dilution of plasmid DNA served as standard for further quantifications. TREC-PB showed 300 copies/µl. Each punch from DBS is equivalent to 3 µl of blood. Assuming 100% extraction efficiency, 900 copies could therefore be expected in 50 µl eluate, resulting in 54 TREC copies per PCR.

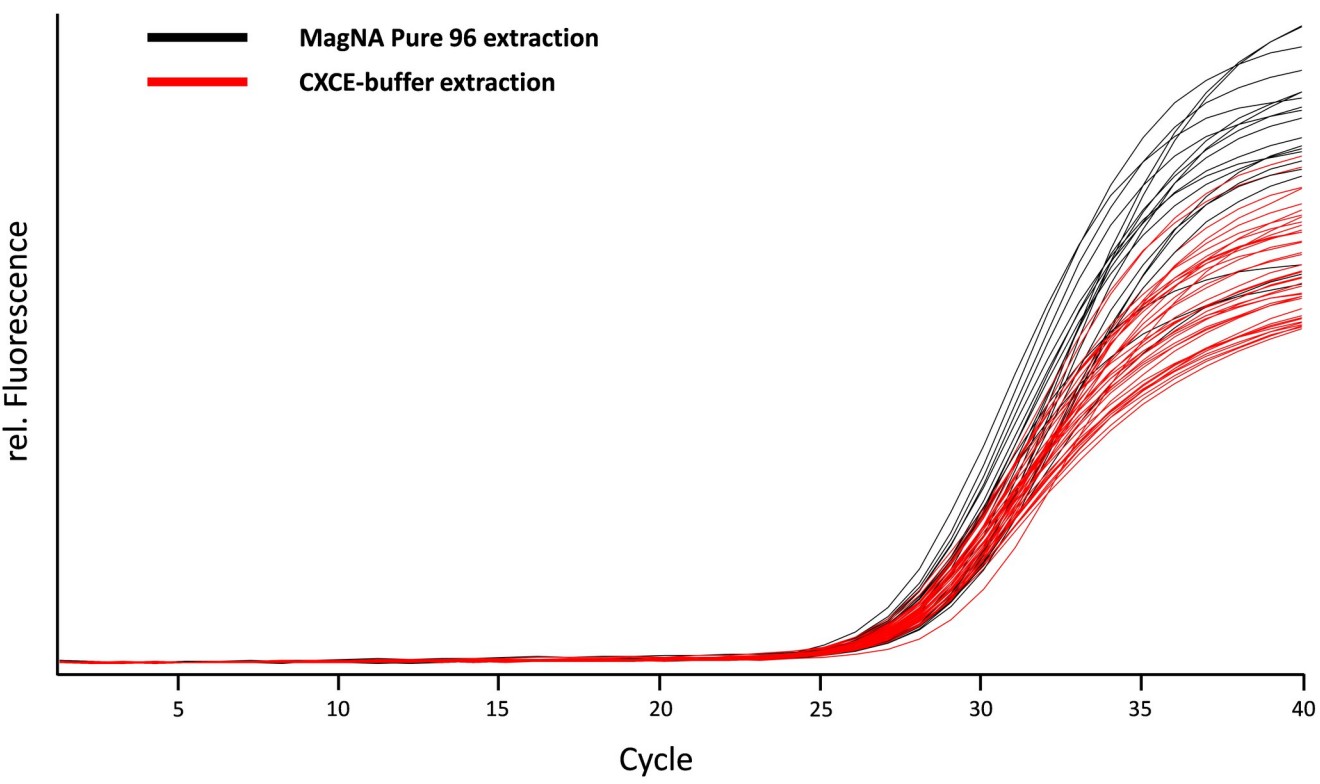

**Fig 2. MagNA pure 96 *vs*. CXCE-buffer extraction.** Comparison of TREC amplification signals from MagNA Pure 96 extraction (black amplification curves) *vs*. CXCE-buffer extraction (red amplification curves).

Therefore, the TREC concentration in our TREC-PB, which mimics a standard healthy newborn, is approximately ten times higher than the assay's detection limit. Dilutions of the TREC-PB could reliably be detected at up to 30 TREC copies/µl, which equals 90 copies per DBS punch (S3 Dataset). These values are nearly identical to those from the plasmid standard.

## Evaluation of pre-analytic DNA carry-over

To analyse the impact of pre-analytic DNA carry-over, TREC negative samples and empty punches were interspersed between TREC and *SMN1* positive samples. We observed Cq values ranging from 27.85 to 30.55 (average 28.86 ± 0.06) with 95% confidence for healthy TREC (PCR positive; TREC-PB) samples. In the TREC-NB and empty punches several contaminations were detected with Cq values ranging from 31.79 to 34.19 (32.91 ± 0.80 standard deviation; Fig 3A, 3B and S4 Dataset). In total, in 10 of 120 wells that were expected to be negative (empty and TREC-NB punches), a TREC signal was detected (Table 1 and S4 Dataset).

For *SMN1*, we observed Cq values ranging from 24.20 to 30.81 (25.5 ± 0.07 with 95% CI, Fig 3C, 3D and S4 dataset) for healthy *SMN1* (PCR positive) samples. In comparison to the TREC assay, more contaminated samples could be detected. Since the *SMN1* gene is also present in the TREC-NB samples, only the empty punches could be considered for evaluation. The *SMN1* signal was present in 32 out of 40 empty punches with Cq values ranging from 31.61 to 35.00 (34.41 ± 0.34 with 95% CI; Fig 3C, 3D and S4 Dataset). Altogether, for *SMN1* and TREC DNA, pre-analytic DNA carry-over was observed in many cases. Therefore, a cut-off had to be established to address the contaminations adequately.

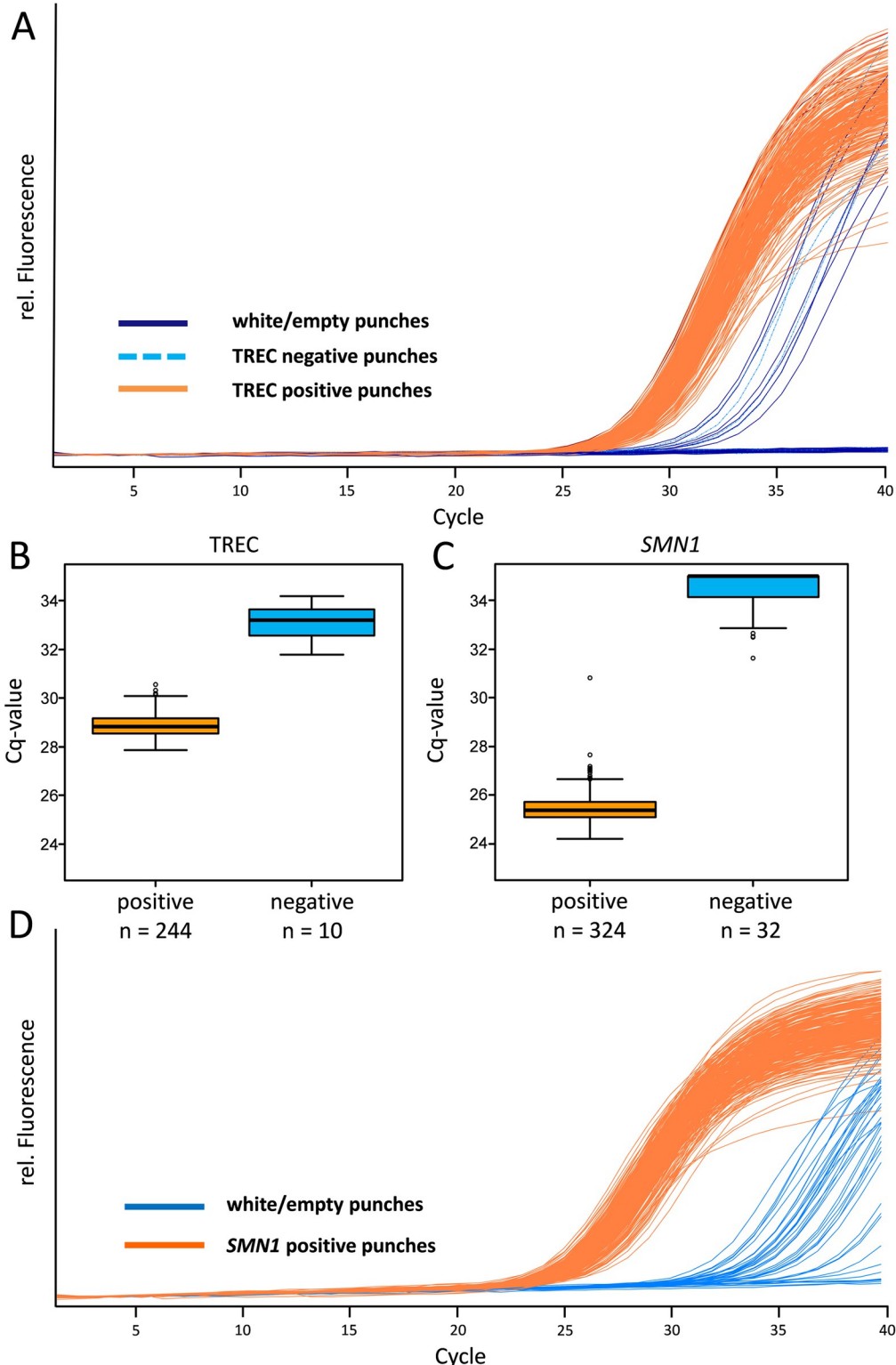

**Fig 3.** Evaluation of pre-analytic DNA carry-over in TREC (A) and *SMN1* (D) amplification curves. In total 364 samples, including 40 white/empty samples, 80 TREC negative samples (dark blue lines: white/empty punches; light blue dashed lines: TREC negative punches; orange lines: positive samples (for TREC≈300 000 copies/ml)). Boxplots and the defined cut-off values based on the Cq values of the TREC (B) and *SMN1* (C) assays with TREC and *SMN1* positive samples in orange, negative samples in blue. Further 110 samples showed no amplification for TREC and 8 no

amplification for *SMN1* and therefore cannot be included in the Boxplot. (D) represents 364 samples, including 40 white/empty (blue lines) and *SMN1* positive samples (orange lines).

**Table 1. Contaminations using the CXCE-buffer extraction.** Results of empty punches embedded between TREC positive samples using the CXCE-buffer extraction and qPCR detection system (Total N = 120).

| Number of samples (n) | TREC status | Percentage |
|---|---|---|
| 110 | negative | 91.67% |
| 10 | TREC amplification signal | 8.33% |

**Table 2. Contaminations using the EnLite system.** Results of empty punches embedded between TREC positive samples using the endpoint PCR-based EnLite system (Total N = 766).

| Number of samples (n) | TREC status | Percentage |
|---|---|---|
| 605 | negative | 79.00% |
| 161 | TREC amplification signal | 21.00% |

The cut-off was determined by subtracting one Cq from the lowest Cq value of the negative samples as fail save (Cq 31.79–1.00 for TREC and Cq 31.61–1.00 for *SMN1*). This value was then rounded to the nearest whole number (Cq 31.00), which is approximately twice the amount of the highest concentration of contaminating DNA detected in the experiments.

In the routine setting all results with a Cq value > 31.00 for the TREC and the *SMN1* assay would therefore be regarded as possible contaminations and retested in duplicates from fresh punches.

The additional TREC copy number measurements of empty punches interspersed between routine samples in Hanover (EnLite system) showed contaminations as well. In total, 161 out of 766 measurements for the negative control punches showed TREC amplification signals (Table 2 and S5 Dataset).

## Unequal distribution of DNA on DBS

DBS on a filter card might show a chromatographic separation pattern therefore punches were taken and analysed from the middle/inner part (area M) and the periphery/edge (area P) of a DBS (Fig 4A and S6 Dataset). Altogether we received positive TREC signals for all samples; neither punches taken from area M nor punches taken from area P were invalid or negative. Amplification curves from the CXCE-buffer extraction were more clustered and showed Cq values between 27.39 and 29.41 for area M, whereas amplification curves from area P showed a broader spread with Cq values varying between 28.06 and 33.07 for TREC (Fig 4B and S6 Dataset). The MagNA Pure extraction showed similar distribution patterns where punches from area M showed Cq values between 26.91 and 28.97 for the TREC fluorescence signal, whereas the punches from area P were detected with Cq values ranging from 28.61 to 31.06. As both isolation methods (CXCE and MagNA Pure 96) do not show significant differences compared to each other, the combined mean Cq value for area M was 28.16 ± 0.10 and 29.98 ± 0.34 for area P with 95% confidence for TREC. These mean Cq values correspond to 101 TREC-copies per PCR reaction for area M and 31 TREC-copies per PCR reaction for area P originating from the same DBS. Those TREC copy numbers per reaction are equivalent to a total number of 560 TREC/µl blood for area M and 170 TREC/µl blood for area P, respectively (Fig 4B and S6 Dataset). The analysis for *SMN1* showed the same distribution pattern

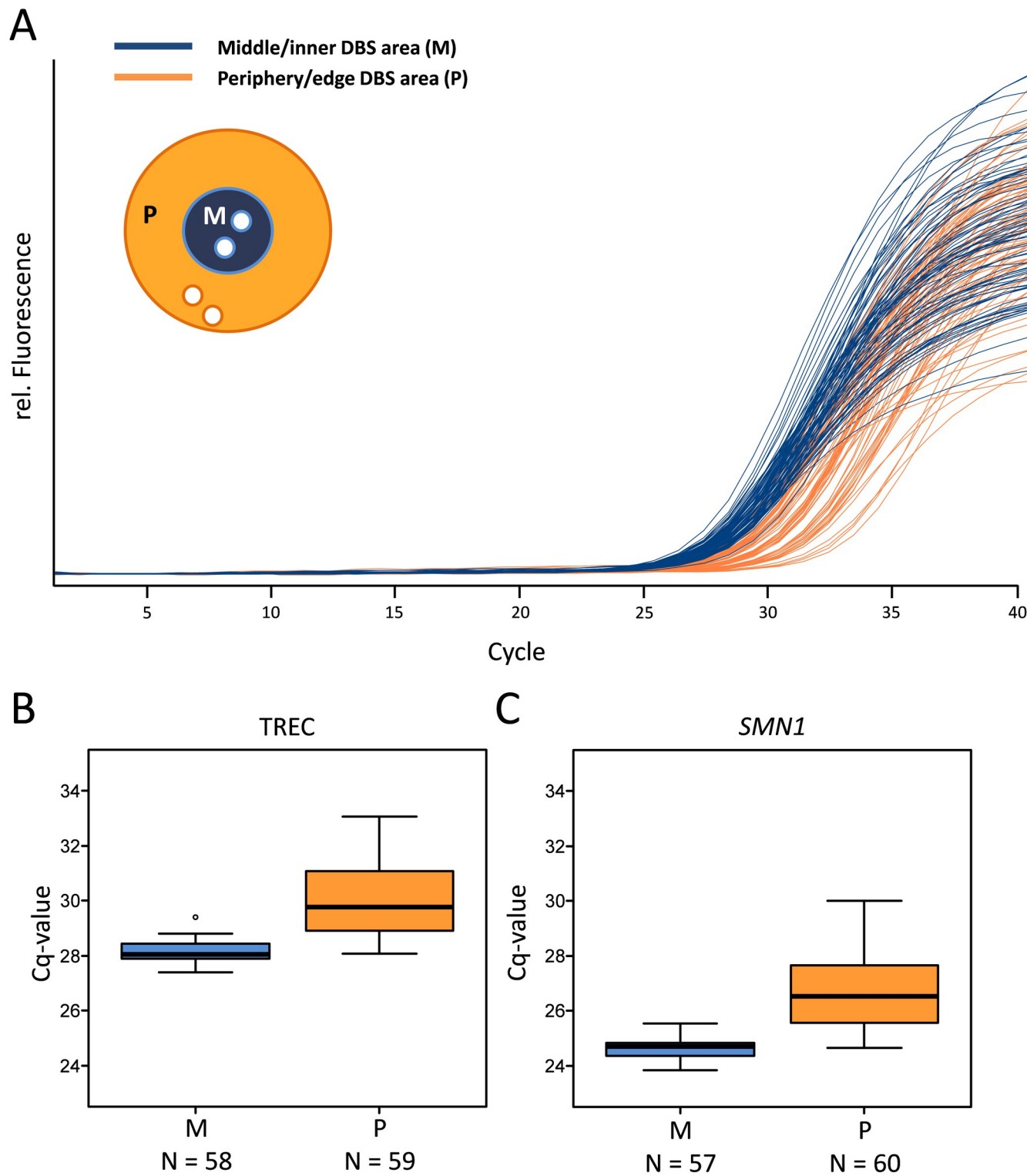

**Fig 4.** Comparison of amplification curves (A) of the middle/inner DBS area (M) *vs.* the periphery/edge DBS area (P) with the CXCE-buffer extraction. The performance of the amplification curves for the nucleic acids of TRECs shows the impact of the chosen punching area within a DBS of the same sample (blue lines/boxes: middle DBS Area (M), orange lines/boxes: periphery DBS area (P)). Direct comparison of the respective Cq values for areas M and P for TREC (B) and *SMN1* (C) DNA.

regarding the punching location, thus punches from area M had more DNA compared to punches from area P (Fig 4C and S6 Dataset). For both, TREC and *SMN1*, the differences between area M and P were statistically significant (p<0.001). Hence, area M punches had significantly more TREC and *SMN1* DNA compared to area P.

## Discussion

SCID can result from variants in numerous genes and is therefore hard to detect on a single gene level. To date, the detection and measurement of TRECs is the only known highly effective and cost-efficient method to identify SCID patients in a high throughput screening [15]. TRECs are episomal DNA circles in recent thymic emigrants; they are stable, do not duplicate during mitosis, and therefore decrease in number with each cell division. Hence, TRECs show a very low copy number compared to genomic DNA [16]. Therefore, in every PCR or qPCR assay, TREC copy numbers in DBS from healthy newborns are already quite close to the detection limit.

Since the statistical probability of a PCR failure increases near the detection limits [30], it is advantageous to increase the TREC copy numbers in the eluate. Using more sample material (e.g. larger or more DBS punches) would not allow to increase copy numbers by more than a factor of two and would cause handling problems. Another possibility would be to improve the extraction efficiency. We therefore evaluated a freeze-thaw step in our CXCE-buffer extraction protocol. Freeze-thaw procedures are commonly used for cell lysis [31]. It has also been shown to be effective in improving DNA or protein extraction from DBS [32]. Indeed, the additional freezing step yielded more TRECs and genomic DNA, probably due to an increased number of disrupted cells. Overall, we achieved a statistically significant shift of 3–7 times higher DNA yield (*SMN1*) and TRECs in the eluate (Fig 1). Due to the improved isolation efficiency, there was less scatter in the Cq values and a statistically significant decrease in Cq values. The implementation of the freezing step led to an improvement in the results of the TREC test, as it produced fewer results with late Cq values. TREC signals with late Cq values that have to be retested were generated without the freezing step. As demonstrated in Fig 1, five samples would have been subject to retesting without freezing, but none with freezing. Taken together, when establishing a novel procedure for SCID screening, the possible effect of a freezing step during or after nucleic acid extraction should be considered.

Importantly, the enhanced CXCE-buffer extraction yielded identical amounts of TREC and genomic DNA as the commercial automated extraction (MagNA Pure 96; Fig 2) and also showed comparable results to the EnLite system for the TREC-PB samples.

The experiments described highlight the major pitfalls associated with using qPCR from DBS in high-throughput neonatal screening for SCID and SMA. In both assays, the absence of a PCR product is a positive screening result and indicates the disease. Therefore, a PCR signal caused by contamination may result in overseeing an affected individual (false-negative screening result).

To analyse the risk of contamination, we examined punches of empty filter cards and TREC-NB samples interspersed between TREC-PB samples. We detected contamination for both SMA and SCID, clearly demonstrating the need for careful evaluation of an appropriate cut-off.

The data demonstrate that TREC has significantly less measurable contamination than SMA. However, the differentiation in the SMA assay is much easier due to the more pronounced difference between healthy individuals and contamination compared to the TREC assay (Fig 3), but a cut-off can be set by using the Cq values of the highest contaminations observed. It can be assumed that the SMA and SCID assays have similar levels of

contamination, but TRECs are not likely to be detected as often as the TREC numbers of healthy individuals are already more close to the detection limit. Taken together, the closer the expected amplification signals are to the detection limit, the more difficult it is to deal with contamination and the more important it is to set a clear cut-off. Contaminations in the CXCE-buffer extraction and LightMix® KIT Newborn assay were observed for 8% of the negative samples (Table 1). Importantly, the comparison with the independent EnLite system also showed similar contaminations. The negative controls in the EnLite assay (endpoint PCR) produced contaminations in 21% of the samples (Table 2). The difference between the Light-Mix® KIT Newborn and the EnLite assay at this point might be due to the fact that the end-point PCR used for the EnLite assay does not utilise amplification curves that could enhance a better separation of results.

Ideally, genetic tests based on PCR can be combined with melting analysis, such as the detection of wildtype and variants in the *HBB* gene for SCD (e.g. LightMix® KIT Newborn). In an assay detecting the wildtype or variant alleles, the problem of contamination is negligible [33]. To change a mutation specific peak to a wildtype peak, high amounts of contaminating wild type DNA would have to be added. Dust particles do not contain enough DNA to cause a contamination resulting in a false (negative) or altered genotype.

In addition, PCR products can easily lead to contaminations. Therefore, it is advisable to establish genetic newborn screening procedures in laboratories and institutions familiar with procedures in molecular biology. This includes, among others, the separation of pre- and post PCR work areas. Another important tool to minimise a risk of contamination with PCR products, is the application of uracil (deoxyuridine triphosphate—dUTP) and uracil N-glycosylase (UNG) in the reaction mix to remove potential contamination by DNA amplicons [19, 20, 34, 35].

Taken together, contaminations play a major role in the interpretation and reliability of the screening analysis. Our data highlight the importance of an appropriate internal laboratory cut-off and an understanding of potential contamination risks and sources. Amplification signals from TREC or *SMN1* alone do not mean that the individual is healthy; especially as the punching process is susceptible to contaminations. There are limited options to reduce this pre-analytic DNA carry-over. Among others, the punchers could be cleaned by punching empty filter cards in between routine samples, however, this might be difficult to implement in a high-throughput setting. Other options could include laser microdissection [36] instead of a punching device, or novel approaches using direct elution without punching [37].

In this study, we were also able to show that the amount of DNA is not evenly distributed across a DBS. Thus, the area (Fig 4) where the punch was taken from the DBS crucially impacted the result. In detail, if samples were punched from the centre (area M) of a DBS significantly more TRECs could be detected as compared to punches from the periphery. This might be due to a physical separation of blood components in the filter paper similar to chromatography. In addition, the process of taking blood samples from newborns is difficult to standardise. Depending on the sampling conditions, different amounts of blood will be dripped onto the filter cards. A relative quantification using a second parameter, such as *SMN1* or a housekeeping gene could be a possible solution for this problem. However, as genomic DNA and TRECs (episomal/ only present in recent thymic emigrants) do not correlate in a specific way, the validity for quantification remains questionable. In addition, acute infections can increase or decrease the total number of T cells and could therefore result in a higher or lower amount of control DNA without increasing the amount of TRECs. Thus, TREC numbers may appear lower or higher compared to the total amount of DNA [38].

A reliable quantification of TRECs, especially when only low numbers can be detected, is desirable for the diagnosis of SCID. However, many different genetic disorders such as

CHARGE or DiGeorge syndromes cause SCID-like clinical signs and are associated with low or absent TREC numbers [39, 40]. Additionally, the process of maturation of the immune system creates further difficulties in SCID screening. Thus, the gestational age plays an important role for false-positive SCID reports. Premature babies, especially those born before 32 weeks gestation, tend to have low or no T cells on their first NBS. This is mainly due to an immature immune system, which typically normalises over time after birth. Regarding the screening, a negative or low TREC signal in a premature newborn does not always mean that the individual has SCID. Most preterm babies achieve a normal TREC count when the screening is repeated a few weeks after birth [41]. In addition, about 40% of SCID patients show maternal T cell engraftment, where maternal T cells migrate across the placenta into the foetus and persist there after birth due to the lack of a functional immune system [42]. This maternal engraftment does not compensate for SCID but might potentially mask the absence of foetal TRECs. All these factors contribute to a high variation of absolute TREC counts. Additionally, a relative quantification is specific for a laboratory and the respective assay applied, so the calculated values are not suitable for comparison between different institutions.

Taken together, our study demonstrates that pre-analytic DNA carry-over is a critical issue for the detection of TRECs in newborn screening and has to be carefully considered when establishing such procedures. Reliable cut-offs to distinguish true PCR signals from contamination for the *SMN1* deletion and TREC can be robustly established by using routine-based setups to assess pre-analytic DNA carry-over. Even the punching location on a DBS plays an important role for the TREC concentration measured, thus an exact quantification of TREC numbers per μl may not be reliable, and therefore should be avoided.

## Conclusion

SMA and SCID (or more correctly TCL) can be reliably screened for. But pre-analytic DNA carry-over remains a major problem in correctly separating negative from positive screening results, especially for TREC-based testing. Therefore, contaminating factors have to be identified, analysed, and validated to reduce the risk of false results by defining the appropriate cut-offs.

To further decrease recall and false-positive rates and characterize potentially affected newborns for SCID/TCL, an epigenetic test to differentiate lymphocyte cell populations may be an interesting tool for future development [43, 44]. Such an epigenetic assay might hold the potential to identify maternal engraftment and therefore reduce the amount of potentially not detectable SCID patients. Also, next generation sequencing might be applied in this context. However, currently, this technique is better suited as a second-tier test, as demonstrated by Fleige *et al.* combining a qPCR screening assay with next generation sequencing [23], or is used for studies [45].

## Supporting information

**S1 Dataset. Raw Cq values for frozen and not frozen samples for the TREC and *SMN1* qPCR analyses.**
(XLSX)

**S2 Dataset. Raw Cq values and calculated concentrations from MagNA Pure 96 and CXCE-buffer extractions of TREC and *SMN1* qPCR analyses.**
(XLSX)

**S3 Dataset. Raw Cq values of dilution series for TREC qPCR analyses.** Datasheets contain (1) the serial dilution of the cloned TIB standard and dilutions of TREC-PB, (2) the results for

the limit of detection using CXCE buffer and (3) the limit of detection raw data for TREC compared for both, the MagNA Pure 96 and CXCE-buffer extractions.
(XLSX)

**S4 Dataset. Raw Cq values of pre-analytic DNA carry-over for TREC and *SMN1*.**
(XLSX)

**S5 Dataset. Raw endpoint PCR counts of pre-analytic DNA carry-over using the EnLite system with the corresponding copy numbers for TREC.**
(XLSX)

**S6 Dataset. Raw Cq values dependent on the area of punching for TREC and *SMN1*.**
(XLSX)

**S1 Fig. Amplification curves of standard serial dilutions for TREC.**
(TIF)

## Author Contributions

**Conceptualization:** Jessica Bzdok, Ludwig Czibere, Siegfried Burggraf, Olfert Landt, Wulf Röschinger, Jürgen Durner.

**Data curation:** Jessica Bzdok, Ludwig Czibere, Siegfried Burggraf, Esther M. Maier, Wulf Röschinger, Michael H. Albert, Sebastian Hegert, Nils Janzen.

**Formal analysis:** Jessica Bzdok, Ludwig Czibere, Siegfried Burggraf, Sebastian Hegert.

**Investigation:** Jessica Bzdok, Ludwig Czibere, Siegfried Burggraf, Marc Becker, Jürgen Durner.

**Methodology:** Jessica Bzdok, Ludwig Czibere, Siegfried Burggraf, Olfert Landt, Esther M. Maier, Wulf Röschinger, Michael H. Albert, Sebastian Hegert, Nils Janzen, Jürgen Durner.

**Project administration:** Jessica Bzdok, Ludwig Czibere, Siegfried Burggraf.

**Resources:** Olfert Landt, Marc Becker, Jürgen Durner.

**Supervision:** Siegfried Burggraf, Esther M. Maier, Wulf Röschinger, Michael H. Albert, Nils Janzen, Marc Becker, Jürgen Durner.

**Validation:** Jessica Bzdok, Ludwig Czibere, Siegfried Burggraf, Olfert Landt.

**Visualization:** Jessica Bzdok, Ludwig Czibere, Siegfried Burggraf.

**Writing – original draft:** Jessica Bzdok, Ludwig Czibere, Siegfried Burggraf, Esther M. Maier, Michael H. Albert.

**Writing – review & editing:** Jessica Bzdok, Ludwig Czibere, Siegfried Burggraf, Olfert Landt, Esther M. Maier, Wulf Röschinger, Michael H. Albert, Sebastian Hegert, Nils Janzen, Marc Becker, Jürgen Durner.

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
