## [Decision Letter · Decision Letter 0]

24 May 2024

PONE-D-24-07899Quality considerations and major pitfalls for high throughput DNA based newborn screening for severe combined immunodeficiency and spinal muscular atrophyPLOS ONE

Dear Dr. Bzdok,

Thank you for submitting your manuscript to PLOS ONE. After careful consideration, we feel that it has merit but does not fully meet PLOS ONE’s publication criteria as it currently stands. Therefore, we invite you to submit a revised version of the manuscript that addresses the points raised during the review process.

We look forward to receiving your revised manuscript.

Kind regards,

Niravkumar Joshi

Academic Editor

PLOS ONE

Journal Requirements:

2. We note that your Data Availability Statement is currently as follows: All relevant data are within the manuscript and its supporting Information files

Reviewers' comments:

Reviewer's Responses to Questions

**Comments to the Author**

1. Is the manuscript technically sound, and do the data support the conclusions?

Reviewer #1: Yes

Reviewer #2: Yes

2. Has the statistical analysis been performed appropriately and rigorously? 

Reviewer #1: Yes

Reviewer #2: No

3. Have the authors made all data underlying the findings in their manuscript fully available?

Reviewer #1: Yes

Reviewer #2: Yes

4. Is the manuscript presented in an intelligible fashion and written in standard English?

Reviewer #1: Yes

Reviewer #2: Yes

5. Review Comments to the Author

**Reviewer #1: **This manuscript focuses on identifying the major drawbacks of current commercial systems for analyzing TREC distribution in the diagnosis of SCID and SMA. The study also proposes a rapid in-house extraction method that is comparable to commercial systems. It was found that carefully considering potential contaminating factors and defining appropriate cut-offs can minimize false results. The experiments were systematically conducted, and the data reasonably supported the conclusions.

A small drawback is that the manuscript contains many abbreviations specific to the field of study. Consequently, every abbreviation, such as PCR, needs to be defined the first time it appears.

Thus, I recommend that the manuscript be accepted, albeit with some small wording changes before publication.

**Reviewer #2:** This manuscript presents a DNA based newborn screening for severe combined immunodeficiency and spinal muscular atrophy. The study holds significant promise, but several revisions are necessary to meet the quality standards of the PLOS ONE journal. Below are specific comments and suggestions for improvement.

1. Authors are advised to change the Italic writing style of Background, Methods, Results, and Conclusions and remove the colon.

2. Combine the introduction content from lines 66-80 on page 4 into a single paragraph. Additionally, remove the numbering for titles throughout the manuscript.

3. Add a Data Collection subsection in the Materials and Methods section, clearly indicating the source of your data.

4. Authors are advised to provide a detailed explanation and discussion regarding the use of freeze-thaw procedures in the Discussion section. Elaborate on how these procedures impact your results and their relevance to the study.

5. Abbreviations need to be defined the first time they appear in the text by writing the full name and giving the abbreviation in brackets.

6. Bold the captions for tables and figures, and change "Figure" to "Fig." as per the formatting guidelines referenced in the articles doi.org/10.1371/journal.pone.0283024;
doi.org/10.1371/journal.pone.0299336.

7. Avoid using bold text within the manuscript for elements such as P and M.

8. Rename the “Perspective” section to “Conclusion” in the manuscript.

9. Bold all the legends in the figures and replace them with high-quality figures.

6. PLOS authors have the option to publish the peer review history of their article (what does this mean?). If published, this will include your full peer review and any attached files.

Reviewer #1: **Yes: **Guangyu Wang

Reviewer #2: No

---

## [Author Response · Author response to Decision Letter 0]

3 Jun 2024

Thank you for the chance of revising our manuscript. This way we can fit it even better to Plos one and the Plos one policies. We also appreciated your suggestions that really helped us to see where we can add significant value to our manuscript and findings. We hope that we could address your points accordingly.

---

## [Decision Letter · Decision Letter 1]

16 Jun 2024

Quality considerations and major pitfalls for high throughput DNA-based newborn screening for severe combined immunodeficiency and spinal muscular atrophy

PONE-D-24-07899R1

Dear Dr. Jessica Bzdok,

We’re pleased to inform you that your manuscript has been judged scientifically suitable for publication and will be formally accepted for publication once it meets all outstanding technical requirements.

Kind regards,

Niravkumar Joshi

Academic Editor

PLOS ONE

Additional Editor Comments (optional):

Reviewers' comments:

Reviewer's Responses to Questions

**Comments to the Author**

1. If the authors have adequately addressed your comments raised in a previous round of review and you feel that this manuscript is now acceptable for publication, you may indicate that here to bypass the “Comments to the Author” section, enter your conflict of interest statement in the “Confidential to Editor” section, and submit your "Accept" recommendation.

Reviewer #1: All comments have been addressed

Reviewer #2: All comments have been addressed

2. Is the manuscript technically sound, and do the data support the conclusions?

Reviewer #1: Yes

Reviewer #2: Yes

3. Has the statistical analysis been performed appropriately and rigorously? 

Reviewer #1: Yes

Reviewer #2: No

4. Have the authors made all data underlying the findings in their manuscript fully available?

Reviewer #1: Yes

Reviewer #2: Yes

5. Is the manuscript presented in an intelligible fashion and written in standard English?

Reviewer #1: Yes

Reviewer #2: Yes

6. Review Comments to the Author

Reviewer #1: The revision has addressed the issues that mentioned in my first review. The authors also made improvements based on the comments of other reviewers.

Reviewer #2: Authors are advised to briefly submit point-by-point comments in their response sheet and highlight the changes in the revised submitted manuscript.

7. PLOS authors have the option to publish the peer review history of their article (what does this mean?). If published, this will include your full peer review and any attached files.

Reviewer #1: **Yes: **Guangyu Wang

Reviewer #2: No

---

## [Editor Report · Acceptance letter]

18 Jun 2024

PONE-D-24-07899R1 

PLOS ONE

Dear Dr. Bzdok, 

I'm pleased to inform you that your manuscript has been deemed suitable for publication in PLOS ONE. Congratulations! Your manuscript is now being handed over to our production team.

Kind regards, 

on behalf of

Dr. Niravkumar Joshi 

Academic Editor

PLOS ONE